# Deep Autoencoding Gaussian Mixture Model for Unsupervised Anomaly Detection

**Bo Zong[†], Qi Song[‡], Martin Renqiang Min[†], Wei Cheng[†]**
**Cristian Lumezanu[†], Daeki Cho[†], Haifeng Chen[†]**
[†]NEC Laboratories America
[‡]Washington State University, Pullman
`{bzong, renqiang, weicheng, lume, dkcho, haifeng}@nec-labs.com`
`qsong@eecs.wsu.edu`

## Abstract

Unsupervised anomaly detection on multi- or high-dimensional data is of great importance in both fundamental machine learning research and industrial applications, for which density estimation lies at the core. Although previous approaches based on dimensionality reduction followed by density estimation have made fruitful progress, they mainly suffer from decoupled model learning with inconsistent optimization goals and incapability of preserving essential information in the low-dimensional space. In this paper, we present a Deep Autoencoding Gaussian Mixture Model (DAGMM) for unsupervised anomaly detection. Our model utilizes a deep autoencoder to generate a low-dimensional representation and reconstruction error for each input data point, which is further fed into a Gaussian Mixture Model (GMM). Instead of using decoupled two-stage training and the standard Expectation-Maximization (EM) algorithm, DAGMM jointly optimizes the parameters of the deep autoencoder and the mixture model simultaneously in an end-to-end fashion, leveraging a separate estimation network to facilitate the parameter learning of the mixture model. The joint optimization, which well balances autoencoding reconstruction, density estimation of latent representation, and regularization, helps the autoencoder escape from less attractive local optima and further reduce reconstruction errors, avoiding the need of pre-training. Experimental results on several public benchmark datasets show that, DAGMM significantly outperforms state-of-the-art anomaly detection techniques, and achieves up to $14\%$ improvement based on the standard $F_1$ score.

## 1 Introduction

Unsupervised anomaly detection is a fundamental problem in machine learning, with critical applications in many areas, such as cybersecurity (Tan et al. (2011)), complex system management (Liu et al. (2008)), medical care (Keller et al. (2012)), and so on. At the core of anomaly detection is density estimation: given a lot of input samples, anomalies are those ones residing in low probability density areas.

Although fruitful progress has been made in the last several years, conducting robust anomaly detection on multi- or high-dimensional data without human supervision remains a challenging task. Especially, when the dimensionality of input data becomes higher, it is more difficult to perform density estimation in the original feature space, as any input sample could be a rare event with low probability to observe (Chandola et al. (2009)). To address this issue caused by the curse of dimensionality, two-step approaches are widely adopted (Candès et al. (2011)), in which dimensionality reduction is first conducted, and then density estimation is performed in the latent low-dimensional space. However, these approaches could easily lead to suboptimal performance, because dimensionality reduction in the first step is unaware of the subsequent density estimation task, and the key information for anomaly detection could be removed in the first place. Therefore, it is desirable to combine the force of dimensionality reduction and density estimation, although a joint optimization accounting for these two components is usually computationally difficult. Several recent

works (Zhai et al. (2016); Yang et al. (2017a); Xie et al. (2016)) explored this direction by utilizing the strong modeling capacity of deep networks, but the resulting performance is limited either by a reduced low-dimensional space that is unable to preserve essential information of input samples, an over-simplified density estimation model without enough capacity, or a training strategy that does not fit density estimation tasks.

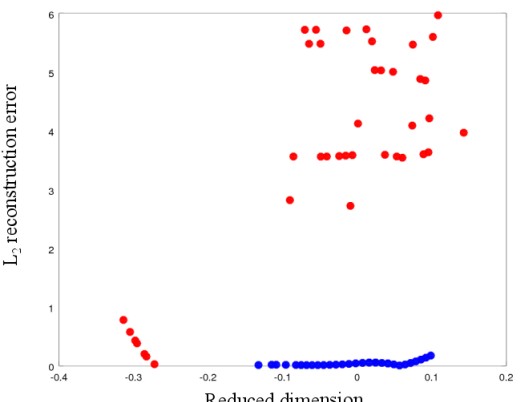

Figure 1: Low-dimensional representations for samples from a private cybersecurity dataset: (1) each sample denotes a network flow that originally has 20 dimensions, (2) red/blue points are abnormal/normal samples, (3) the horizontal axis denotes the reduced 1-dimensional space learned by a deep autoencoder, and (4) the vertical axis denotes the reconstruction error induced by the 1-dimensional representation.

In this paper, we propose Deep Autoencoding Gaussian Mixture Model (DAGMM), a deep learning framework that addresses the aforementioned challenges in unsupervised anomaly detection from several aspects.

First, DAGMM preserves the key information of an input sample in a low-dimensional space that includes features from both the reduced dimensions discovered by dimensionality reduction and the induced reconstruction error. From the example shown in Figure 1, we can see that anomalies differ from normal samples in two aspects: (1) anomalies can be significantly deviated in the reduced dimensions where their features are correlated in a different way; and (2) anomalies are harder to reconstruct, compared with normal samples. Unlike existing methods that only involve one of the aspects (Zimek et al. (2012); Zhai et al. (2016)) with sub-optimal performance, DAGMM utilizes a sub-network called compression network to perform dimensionality reduction by an autoencoder, which prepares a low-dimensional representation for an input sample by concatenating reduced low-dimensional features from encoding and the reconstruction error from decoding.

Second, DAGMM leverages a Gaussian Mixture Model (GMM) over the learned low-dimensional space to deal with density estimation tasks for input data with complex structures, which are yet rather difficult for simple models used in existing works (Zhai et al. (2016)). While GMM has strong capability, it also introduces new challenges in model learning. As GMM is usually learned by alternating algorithms such as Expectation-Maximization (EM) (Huber (2011)), it is hard to perform joint optimization of dimensionality reduction and density estimation favoring GMM learning, which is often degenerated into a conventional two-step approach. To address this training challenge, DAGMM utilizes a sub-network called estimation network that takes the low-dimensional input from the compression network and outputs mixture membership prediction for each sample. With the predicted sample membership, we can directly estimate the parameters of GMM, facilitating the evaluation of the energy/likelihood of input samples. By simultaneously minimizing reconstruction error from compression network and sample energy from estimation network, we can jointly train a dimensionality reduction component that directly helps the targeted density estimation task.

Finally, DAGMM is friendly to end-to-end training. Usually, it is hard to learn deep autoencoders by end-to-end training, as they can be easily stuck in less attractive local optima, so pre-training is widely adopted (Vincent et al. (2010); Yang et al. (2017a); Xie et al. (2016)). However, pre-training limits the potential to adjust the dimensionality reduction behavior because it is hard to make any

significant change to a well-trained autoencoder via fine-tuning. Our empirical study demonstrates that, DAGMM is well-learned by the end-to-end training, as the regularization introduced by the estimation network greatly helps the autoencoder in the compression network escape from less attractive local optima.

Experiments on several public benchmark datasets demonstrate that, DAGMM has superior performance over state-of-the-art techniques, with up to $14\%$ improvement of F1 score for anomaly detection. Moreover, we observe that the reconstruction error from the autoencoder in DAGMM by the end-to-end training is as low as the one made by its pre-trained counterpart, while the reconstruction error from an autoencoder without the regularization from the estimation network stays high. In addition, the end-to-end trained DAGMM significantly outperforms all the baseline methods that rely on pre-trained autoencoders.

## 2 RELATED WORK

Tremendous effort has been devoted to unsupervised anomaly detection (Chandola et al. (2009)), and the existing methods can be grouped into three categories.

Reconstruction based methods assume that anomalies are incompressible and thus cannot be effectively reconstructed from low-dimensional projections. Conventional methods in this category include Principal Component Analysis (PCA) (Jolliffe (1986)) with explicit linear projections, kernel PCA with implicit non-linear projections induced by specific kernels (Günter et al.), and Robust PCA (RPCA) (Huber (2011); Candès et al. (2011)) that makes PCA less sensitive to noise by enforcing sparse structures. In addition, multiple recent works propose to analyze the reconstruction error induced by deep autoencoders, and demonstrate promising results (Zhou & Paffenroth (2017); Zhai et al. (2016)). However, the performance of reconstruction based methods is limited by the fact that they only conduct anomaly analysis from a single aspect, that is, reconstruction error. Although the compression on anomalous samples could be different from the compression on normal samples and some of them do demonstrate unusually high reconstruction errors, a significant amount of anomalous samples could also lurk with a normal level of error, which usually happens when the underlying dimensionality reduction methods have high model complexity or the samples of interest are noisy with complex structures. Even in these cases, we still have the hope to detect such "lurking" anomalies, as they still reside in low-density areas in the reduced low-dimensional space. Unlike the existing reconstruction based methods, DAGMM considers the both aspects, and performs density estimation in a low-dimensional space derived from the reduced representation and the reconstruction error caused by the dimensionality reduction, for a comprehensive view.

Clustering analysis is another popular category of methods used for density estimation and anomaly detection, such as multivariate Gaussian Models, Gaussian Mixture Models, $k$-means, and so on (Barnett & Lewis (1984); Zimek et al. (2012); Kim & Scott (2011); Xiong et al. (2011)). Because of the curse of dimensionality, it is difficult to directly apply such methods to multi- or high- dimensional data. Traditional techniques adopt a two-step approach (Chandola et al. (2009)), where dimensionality reduction is conducted first, then clustering analysis is performed, and the two steps are separately learned. One of the drawbacks in the two-step approach is that dimensionality reduction is trained without the guidance from the subsequent clustering analysis, thus the key information for clustering analysis could be lost during dimensionality reduction. To address this issue, recent works propose deep autoencoder based methods in order to jointly learn dimensionality reduction and clustering components. However, the performance of the state-of-the-art methods is limited by over-simplified clustering models that are unable to handle clustering or density estimation tasks for data of complex structures, or the pre-trained dimensionality reduction component (*i.e.,* autoencoder) has little potential to accommodate further adjustment by the subsequent fine-tuning for anomaly detection. DAGMM explicitly addresses these issues by a sub-network called estimation network that evaluates sample density in the low-dimensional space produced by its compression network. By predicting sample mixture membership, we are able to estimate the parameters of GMM without EM-like alternating procedures. Moreover, DAGMM is friendly to end-to-end training so that we can unleash the full potential of adjusting dimensionality reduction components and jointly improve the quality of clustering analysis/density estimation.

In addition, one-class classification approaches are also widely used for anomaly detection. Under this framework, a discriminative boundary surrounding the normal instances is learned by algorithms,

such as one-class SVM (Chen et al. (2001); Song et al. (2002); Williams et al. (2002)). When the number of dimensions grows higher, such techniques usually suffer from suboptimal performance due to the curse of dimensionality. Unlike these methods, DAGMM estimates data density in a jointly learned low-dimensional space for more robust anomaly detection.

There has been growing interest in joint learning of dimensionality reduction (feature selection) and Gaussian mixture modeling. Yang et al. (2014; 2017b) propose a method that jointly learns linear dimensionality reduction and GMM. Paulik (2013) studies how to perform better feature selection with a pre-trained GMM as a regularizer. Variani et al. (2015) and Zhang & Woodland (2017) propose joint learning frameworks, where the parameters of GMM are directly estimated through supervision information in speech recognition applications. Tüske et al. (2015a;b) investigate how to use log-linear mixture models to approximate GMM posterior under the conditions that a class/mixture prior distribution is given and a covariance matrix is globally shared. Unlike the existing works, we focus on unsupervised settings: DAGMM extracts useful features for anomaly detection through non-linear dimensionality reduction realized by a deep autoencoder, and jointly learns their density under the GMM framework by mixture membership estimation, for which DAGMM can be viewed as a more powerful deep unsupervised version of adaptive mixture of experts (Jacobs et al. (1991)) in combination with a deep autoencoder. More importantly, DAGMM combines induced reconstruction error and learned latent representation for unsupervised anomaly detection.

## 3 DEEP AUTOENCODING GAUSSIAN MIXTURE MODEL

### 3.1 OVERVIEW

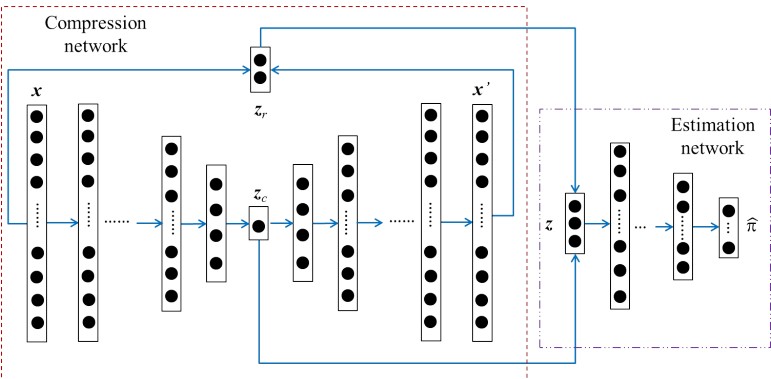

Figure 2: An overview on Deep Autoencoding Gaussian Mixture Model

Deep Autoencoding Gaussian Mixture Model (DAGMM) consists of two major components: a compression network and an estimation network. As shown in Figure 2, DAGMM works as follows: (1) the compression network performs dimensionality reduction for input samples by a deep autoencoder, prepares their low-dimensional representations from both the reduced space and the reconstruction error features, and feeds the representations to the subsequent estimation network; (2) the estimation network takes the feed, and predicts their likelihood/energy in the framework of Gaussian Mixture Model (GMM).

### 3.2 COMPRESSION NETWORK

The low-dimensional representations provided by the compression network contains two sources of features: (1) the reduced low-dimensional representations learned by a deep autoencoder; and (2) the features derived from reconstruction error. Given a sample $\mathbf{x}$, the compression network computes its low-dimensional representation $\mathbf{z}$ as follows.

$$\mathbf{z}_c = h(\mathbf{x}; \theta_e), \qquad \qquad \mathbf{x}' = g(\mathbf{z}_c; \theta_d), \qquad (1)$$

$$\mathbf{z}_r = f(\mathbf{x}, \mathbf{x}'), \qquad (2)$$

$$\mathbf{z} = [\mathbf{z}_c, \mathbf{z}_r], \qquad (3)$$

where $\mathbf{z}_c$ is the reduced low-dimensional representation learned by the deep autoencoder, $\mathbf{z}_r$ includes the features derived from the reconstruction error, $\theta_e$ and $\theta_d$ are the parameters of the deep autoencoder, $\mathbf{x}'$ is the reconstructed counterpart of $\mathbf{x}$, $h(\cdot)$ denotes the encoding function, $g(\cdot)$ denotes the decoding function, and $f(\cdot)$ denotes the function of calculating reconstruction error features. In particular, $\mathbf{z}_r$ can be multi-dimensional, considering multiple distance metrics such as absolute Euclidean distance, relative Euclidean distance, cosine similarity, and so on. In the end, the compression network feeds $\mathbf{z}$ to the subsequent estimation network.

## 3.3 ESTIMATION NETWORK

Given the low-dimensional representations for input samples, the estimation network performs density estimation under the framework of GMM.

In the training phase with unknown mixture-component distribution $\phi$, mixture means $\mu$, and mixture covariance $\mathbf{\Sigma}$, the estimation network estimates the parameters of GMM and evaluates the likelihood/energy for samples without alternating procedures such as EM (Zimek et al. (2012)). The estimation network achieves this by utilizing a multi-layer neural network to predict the mixture membership for each sample. Given the low-dimensional representations $\mathbf{z}$ and an integer $K$ as the number of mixture components, the estimation network makes membership prediction as follows.

$$\mathbf{p} = MLN(\mathbf{z}; \theta_m), \qquad\qquad \hat{\gamma} = \text{softmax}(\mathbf{p}), \qquad\qquad (4)$$

where $\hat{\gamma}$ is a $K$-dimensional vector for the soft mixture-component membership prediction, and $\mathbf{p}$ is the output of a multi-layer network parameterized by $\theta_m$. Given a batch of $N$ samples and their membership prediction, $\forall 1 \leq k \leq K$, we can further estimate the parameters in GMM as follows.

$$\hat{\phi}_k = \sum_{i=1}^{N} \frac{\hat{\gamma}_{ik}}{N}, \qquad \hat{\mu}_k = \frac{\sum_{i=1}^{N} \hat{\gamma}_{ik} \mathbf{z}_i}{\sum_{i=1}^{N} \hat{\gamma}_{ik}}, \qquad \hat{\mathbf{\Sigma}}_k = \frac{\sum_{i=1}^{N} \hat{\gamma}_{ik}(\mathbf{z}_i - \hat{\mu}_k)(\mathbf{z}_i - \hat{\mu}_k)^T}{\sum_{i=1}^{N} \hat{\gamma}_{ik}}. \qquad (5)$$

where $\hat{\gamma}_i$ is the membership prediction for the low-dimensional representation $\mathbf{z}_i$, and $\hat{\phi}_k$, $\hat{\mu}_k$, $\hat{\mathbf{\Sigma}}_k$ are mixture probability, mean, covariance for component $k$ in GMM, respectively.

With the estimated parameters, sample energy can be further inferred by

$$E(\mathbf{z}) = -\log \left( \sum_{k=1}^{K} \hat{\phi}_k \frac{\exp\left(-\frac{1}{2}(\mathbf{z} - \hat{\mu}_k)^T \hat{\mathbf{\Sigma}}_k^{-1}(\mathbf{z} - \hat{\mu}_k)\right)}{\sqrt{|2\pi\hat{\mathbf{\Sigma}}_k|}} \right). \qquad (6)$$

where $|\cdot|$ denotes the determinant of a matrix.

In addition, during the testing phase with the learned GMM parameters, it is straightforward to estimate sample energy, and predict samples of high energy as anomalies by a pre-chosen threshold.

## 3.4 OBJECTIVE FUNCTION

Given a dataset of $N$ samples, the objective function that guides DAGMM training is constructed as follows.

$$J(\theta_e, \theta_d, \theta_m) = \frac{1}{N} \sum_{i=1}^{N} L(\mathbf{x}_i, \mathbf{x}'_i) + \frac{\lambda_1}{N} \sum_{i=1}^{N} E(\mathbf{z}_i) + \lambda_2 P(\hat{\mathbf{\Sigma}}). \qquad (7)$$

This objective function includes three components.

- $L(\mathbf{x}_i, \mathbf{x}'_i)$ is the loss function that characterizes the reconstruction error caused by the deep autoencoder in the compression network. Intuitively, if the compression network could make the reconstruction error low, the low-dimensional representation could better preserve the key information of input samples. Therefore, a compression network of lower reconstruction error is always desired. In practice, $L_2$-norm usually gives desirable results, as $L(\mathbf{x}_i, \mathbf{x}'_i) = \|\mathbf{x}_i - \mathbf{x}'_i\|_2^2$.
- $E(\mathbf{z}_i)$ models the probabilities that we could observe the input samples. By minimizing the sample energy, we look for the best combination of compression and estimation networks that maximize the likelihood to observe input samples.

- DAGMM also has the singularity problem as in GMM: trivial solutions are triggered when the diagonal entries in covariance matrices degenerate to 0. To avoid this issue, we penalize small values on the diagonal entries by $P(\hat{\boldsymbol{\Sigma}}) = \sum_{k=1}^{K} \sum_{j=1}^{d} \frac{1}{\hat{\boldsymbol{\Sigma}}_{kjj}}$, where $d$ is the number of dimensions in the low-dimensional representations provided by the compression network.
- $\lambda_1$ and $\lambda_2$ are the meta parameters in DAGMM. In practice, $\lambda_1 = 0.1$ and $\lambda_2 = 0.005$ usually render desirable results.

## 3.5 Relation to Variational Inference

In DAGMM, we leverage the estimation network to make membership prediction for each sample. From the view of probabilistic graphical models, the estimation network plays an analogous role of latent variable (*i.e.,* sample membership) inference. Recently, neural variational inference (Mnih & Gregor (2014)) has been proposed to employ deep neural networks to tackle difficult latent variable inference problems, where exact model inference is intractable and conventional approximate methods cannot scale well. Theoretically, we can also adapt the membership prediction task of DAGMM into the framework of neural variational inference. For sample $\mathbf{x}_i$, the contribution of its compressed representation $\mathbf{z}_i$ to the energy function can be upper-bounded as follows (Jordan et al. (1999)),

$$E(\mathbf{z}_i) = -\log p(\mathbf{z}_i) = -\log \sum_k p(\mathbf{z}_i, k)$$

$$= -\log \sum_k Q_{\theta_m}(k \mid \mathbf{z}_i) \frac{p(\mathbf{z}_i, k)}{Q_{\theta_m}(k \mid \mathbf{z}_i)}$$

$$\leq -\sum_k Q_{\theta_m}(k \mid \mathbf{z}_i) \log \frac{p(\mathbf{z}_i, k)}{Q_{\theta_m}(k \mid \mathbf{z}_i)}$$

$$= -E_{Q_{\theta_m}}[\log p(\mathbf{z}_i, k) - \log Q_{\theta_m}(k \mid \mathbf{z}_i)] \tag{8}$$

$$= -E_{Q_{\theta_m}}[\log p(\mathbf{z}_i \mid k)] + \mathrm{KL}(Q_{\theta_m}(k \mid \mathbf{z}_i) \| p(k)) \tag{9}$$

$$= -\log p(\mathbf{z}_i) + \mathrm{KL}(Q_{\theta_m}(k \mid \mathbf{z}_i) \| p(k \mid \mathbf{z}_i))$$

$$= E(\mathbf{z}_i) + \mathrm{KL}(Q_{\theta_m}(k \mid \mathbf{z}_i) \| p(k \mid \mathbf{z}_i)) \tag{10}$$

where $Q_{\theta_m}(k \mid \mathbf{z}_i)$ is the estimation network that predicts the membership of $\mathbf{z}_i$, $\mathrm{KL}(\cdot \| \cdot)$ is the Kullback-Leibler divergence between two input distributions, $p(k) = \phi_k$ is the mixing coefficient to be estimated, and $p(k \mid \mathbf{z}_i)$ is the posterior probability distribution of mixture component $k$ given $\mathbf{z}_i$.

By minimizing the negative evidence lower bound in Equation (8), we can make the estimation network approximate the true posterior and tighten the bound of energy function. In DAGMM, we use Equation (6) as a part of the objective function instead of its upper bound in Equation (10) simply because the energy function of DAGMM is tractable and efficient to evaluate. Unlike neural variational inference that uses the deep estimation network to define a variational posterior distribution as described above, DAGMM explicitly employs the deep estimation network to parametrize a sample-dependent prior distribution. In the history of machine learning research, there were research efforts towards utilizing neural networks to calculate sample membership in mixture models, such as adaptive mixture of experts (Jacobs et al. (1991)). From this perspective, DAGMM can be viewed as a powerful deep unsupervised version of adaptive mixture of experts in combination with a deep autoencoder.

## 3.6 Training Strategy

Unlike existing deep autoencoder based methods (Yang et al. (2017a); Xie et al. (2016)) that rely on pre-training, DAGMM employs end-to-end training. First, in our study, we find that pre-trained compression networks suffer from limited anomaly detection performance, as it is difficult to make significant changes in the well-trained deep autoencoder to favor the subsequent density estimation tasks. Second, we also find that the compression network and estimation network could mutually boost each others' performance. On one hand, with the regularization introduced by the estimation network, the deep autoencoder in the compression network learned by end-to-end training can reduce reconstruction error as low as the error from its pre-trained counterpart, which meanwhile cannot be achieved by simply performing end-to-end training with the deep autoencoder alone. On the other hand, with the well-learned low-dimensional representations from the compression network, the estimation network is able to make meaningful density estimations.

In Section 4.5, we employ an example from a public benchmark dataset to discuss the choice between pre-training and end-to-end training in DAGMM.

# 4 EXPERIMENTAL RESULTS

In this section, we use public benchmark datasets to demonstrate the effectiveness of DAGMM in unsupervised anomaly detection.

## 4.1 DATASET

|  | # Dimensions | # Instances | Anomaly ratio ($\rho$) |
|---|---|---|---|
| KDDCUP | 120 | 494,021 | 0.2 |
| Thyroid | 6 | 3,772 | 0.025 |
| Arrhythmia | 274 | 452 | 0.15 |
| KDDCUP-Rev | 120 | 121,597 | 0.2 |

Table 1: Statistics of the public benchmark datasets

We employ four benchmark datasets: KDDCUP, Thyroid, Arrhythmia, and KDDCUP-Rev.

- **KDDCUP**. The KDDCUP99 10 percent dataset from the UCI repository (Lichman (2013)) originally contains samples of 41 dimensions, where 34 of them are continuous and 7 are categorical. For categorical features, we further use one-hot representation to encode them, and eventually we obtain a dataset of 120 dimensions. As 20% of data samples are labeled as "normal" and the rest are labeled as "attack", "normal" samples are in a minority group; therefore, "normal" ones are treated as anomalies in this task.

- **Thyroid**. The Thyroid (Lichman (2013)) dataset is obtained from the ODDS repository [1]. There are 3 classes in the original dataset. In this task, the hyperfunction class is treated as anomaly class and the other two classes are treated as normal class, because hyperfunction is a clear minority class.

- **Arrhythmia**. The Arrhythmia (Lichman (2013)) dataset is also obtained from the ODDS repository. The smallest classes, including 3, 4, 5, 7, 8, 9, 14, and 15, are combined to form the anomaly class, and the rest of the classes are combined to form the normal class.

- **KDDCUP-Rev**. This dataset is derived from KDDCUP. We keep all the data samples labeled as "normal" and randomly draw samples labeled as "attack" so that the ratio between "normal" and "attack" is $4 : 1$. In this way, we obtain a dataset with anomaly ratio 0.2, where "attack" samples are in a minority group and treated as anomalies. Note that "attack" samples are not fixed, and we randomly draw "attack" samples in every single run.

Detailed information about the datasets is shown in Table 1.

## 4.2 BASELINE METHODS

We consider both traditional and state-of-the-art deep learning methods as baselines.

- OC-SVM. One-class support vector machine (Chen et al. (2001)) is a popular kernel-based method used in anomaly detection. In the experiment, we employ the widely adopted radial basis function (RBF) kernel in all the tasks.

- DSEBM-e. Deep structured energy based model (DSEBM) (Zhai et al. (2016)) is a state-of-the-art deep learning method for unsupervised anomaly detection. In DSEBM-e, sample energy is leveraged as the criterion to detect anomalies.

- DSEBM-r. DSEBM-e and DSEBM-r (Zhai et al. (2016)) share the same core technique, but reconstruction error is used as the criterion in DSEBM-r for anomaly detection.

---

[1]http://odds.cs.stonybrook.edu/

- DCN. Deep clustering network (DCN) (Yang et al. (2017a)) is a state-of-the-art clustering algorithm that regulates autoencoder performance by k-means. We adapt this technique to anomaly detection tasks. In particular, the distance between a sample and its cluster center is taken as the criterion for anomaly detection: samples that are farther from their cluster centers are more likely to be anomalies.

Moreover, we include the following DAGMM variants as baselines to demonstrate the importance of individual components in DAGMM.

- GMM-EN. In this variant, we remove the reconstruction error component from the objective function of DAGMM. In other words, the estimation network in DAGMM performs membership estimation without the constraints from the compression network. With the learned membership estimation, we infer sample energy by Equation (5) and (6) under the GMM framework. Sample energy is used as the criterion for anomaly detection.

- PAE. We obtain this variant by removing the energy function from the objective function of DAGMM, and this DAGMM variant is equivalent to a deep autoenoder. To ensure the compression network is well trained, we adopt the pre-training strategy (Vincent et al. (2010)). Sample reconstruction error is the criterion for anomaly detection.

- E2E-AE. This variant shares the same setting with PAE, but the deep autoencoder is learned by end-to-end training. Sample reconstruction error is the criterion for anomaly detection

- PAE-GMM-EM. This variant adopts a two-step approach. At step one, we learn the compression network by pre-training deep autoencoder. At step two, we use the output from the compression network to train the GMM by a traditional EM algorithm. The training procedures in the two steps are separated. Sample energy is used as the criterion for anomaly detection.

- PAE-GMM. This variant also adopts a two-step approach. At step one, we learn the compression network by pre-training deep autoencoder. At step two, we use the output from the compression network to train the estimation network. The training procedures in the two steps are separated. Sample energy is used as the criterion for anomaly detection.

- DAGMM-p. This variant is a compromise between DAGMM and PAE-GMM: we first train the compression network by pre-training, and then fine-tune DAGMM by end-to-end training. Sample energy is the criterion for anomaly detection.

- DAGMM-NVI. The only difference between this variant and DAGMM is that this variant adopts the framework of neural variational inference (Mnih & Gregor (2014)) and replaces Equation (6) with the upper bound in Equation (10) as a part of the objective function.

## 4.3 DAGMM Configuration

In all the experiment, we consider two reconstruction features from the compression network: relative Euclidean distance and cosine similarity. Given a sample $\mathbf{x}$ and its reconstructed counterpart $\mathbf{x}'$, their relative Euclidean distance is defined as $\frac{\|\mathbf{x} - \mathbf{x}'\|_2}{\|\mathbf{x}\|_2}$, and the cosine similarity is derived by $\frac{\mathbf{x} \cdot \mathbf{x}'}{\|\mathbf{x}\|_2 \|\mathbf{x}'\|_2}$. In Appendix D, for readers of interest, we discuss why reconstruction features are important to DAGMM and how to select reconstruction features in practice.

The network structures of DAGMM used on individual datasets are summarized as follows.

- KDDCUP. For this dataset, its compression network provides 3 dimensional input to the estimation network, where one is the reduced dimension and the other two are from the reconstruction error. The estimation network considers a GMM with 4 mixture components for the best performance. In particular, the compression network runs with FC(120, 60, $\tanh$)-FC(60, 30, $\tanh$)-FC(30, 10, $\tanh$)-FC(10, 1, none)-FC(1, 10, $\tanh$)-FC(10, 30, $\tanh$)-FC(30, 60, $\tanh$)-FC(60, 120, none), and the estimation network performs with FC(3, 10, $\tanh$)-Drop(0.5)-FC(10, 4, softmax).

- Thyroid. The compression network for this dataset also provides 3 dimensional input to the estimation network, and the estimation network employs 2 mixture components for the best performance. In particular, the compression network runs with FC(6, 12, $\tanh$)-FC(12, 4,

tanh)-FC(4, 1, none)-FC(1, 4, tanh)-FC(4, 12, tanh)-FC(12, 6, none), and the estimation network performs with FC(3, 10, tanh)-Drop(0.5)-FC(10, 2, softmax).

- Arrhythmia. The compression network for this dataset provides 4 dimensional input, where two of them are the reduced dimensions, and the estimation network adopts a setting of 2 mixture components for the best performance. In particular, the compression network runs with FC(274, 10, tanh)-FC(10, 2, none)-FC(2, 10, tanh)-FC(10, 274, none), and the estimation network performs with FC(4, 10, tanh)-Drop(0.5)-FC(10, 2, softmax).

- KDDCUP-Rev. For this dataset, its compression network provides 3 dimensional input to the estimation network, where one is the reduced dimension and the other two are from the reconstruction error. The estimation network considers a GMM with 2 mixture components for the best performance. In particular, the compression network runs with FC(120, 60, tanh)-FC(60, 30, tanh)-FC(30, 10, tanh)-FC(10, 1, none)-FC(1, 10, tanh)-FC(10, 30, tanh)-FC(30, 60, tanh)-FC(60, 120, none), and the estimation network performs with FC(3, 10, tanh)-Drop(0.5)-FC(10, 2, softmax).

where FC($a$, $b$, $f$) means a fully-connected layer with $a$ input neurons and $b$ output neurons activated by function $f$ (none means no activation function is used), and Drop($p$) denotes a dropout layer with keep probability $p$ during training.

All the DAGMM instances are implemented by tensorflow (Abadi et al. (2016)) and trained by Adam (Kingma & Ba (2015)) algorithm with learning rate 0.0001. For KDDCUP, Thyroid, Arrhythmia, and KDDCUP-Rev, the number of training epochs are 200, 20000, 10000, and 400, respectively. For the sizes of mini-batches, they are set as 1024, 1024, 128, and 1024, respectively. Moreover, in all the DAGMM instances, we set $\lambda_1$ as 0.1 and $\lambda_2$ as 0.005. For readers of interest, we discuss how $\lambda_1$ and $\lambda_2$ impact DAGMM in Appendix F.

For the baseline methods, we conduct exhaustive search to find the optimal meta parameters for them in order to achieve the best performance. We detail their exact configuration in Appendix A.

## 4.4 Accuracy

**Metric**. We consider average precision, recall, and $F_1$ score as intuitive ways to compare anomaly detection performance. In particular, based on the anomaly ratio suggested in Table 1, we select the threshold to identify anomalous samples. For example, when DAGMM performs on KDDCUP, the top 20% samples of the highest energy will be marked as anomalies. We take anomaly class as positive, and define precision, recall, and $F_1$ score accordingly.

In the first set of experiment, we follow the setting in (Zhai et al. (2016)) with completely clean training data: in each run, we take 50% of data by random sampling for training with the rest 50% reserved for testing, and only data samples from the normal class are used for training models.

Table 2 reports the average precision, recall, and $F_1$ score after 20 runs for DAGMM and its baselines. In general, DAGMM demonstrates superior performance over the baseline methods in terms of $F_1$ score on all the datasets. Especially on KDDCUP and KDDCUP-Rev, DAGMM achieves 14% and 10% improvement at $F_1$ score, compared with the existing methods. For OC-SVM, the curse of dimensionality could be the main reason that limits its performance. For DSEBM, while it works reasonably well on multiple datasets, DAGMM outperforms as both latent representation and reconstruction error are jointly considered in energy modeling. For DCN, PAE-GMM, and DAGMM-p, their performance could be limited by the pre-trained deep autoencoders. When a deep autoencoder is well-trained, it is hard to make any significant change on the reduced dimensions and favor the subsequent density estimation tasks. For GMM-EN, without the reconstruction constraints, it seems difficult to perform reasonable density estimation. In terms of PAE, the single view of reconstruction error may not be sufficient for anomaly detection tasks. For E2E-AE, we observe that it is unable to reduce reconstruction error as low as PAE and DAGMM do on KDDCUP, KDDCUP-Rev, and Thyroid. As the key information of data could be lost during dimensionality reduction, E2E-AE suffers poor performance on KDDCUP and Thyroid. In addition, the performance of DAGMM and DAGMM-NVI is quite similar. As GMM is a fairly simple graphical model, we cannot spot significant improvement brought by neural variational inference in DAGMM. In Appendix B, for readers of interest, we show the cumulative distribution functions of the energy function learned by DAGMM for all the datasets under the setting of clean training data.

| Method | KDDCUP | | | Thyroid | | |
|---|---|---|---|---|---|---|
| | Precision | Recall | $F_1$ | Precision | Recall | $F_1$ |
| OC-SVM | 0.7457 | 0.8523 | 0.7954 | 0.3639 | 0.4239 | 0.3887 |
| DSEBM-r | 0.1972 | 0.2001 | 0.1987 | 0.0404 | 0.0403 | 0.0403 |
| DSEBM-e | 0.7369 | 0.7477 | 0.7423 | 0.1319 | 0.1319 | 0.1319 |
| DCN | 0.7696 | 0.7829 | 0.7762 | 0.3319 | 0.3196 | 0.3251 |
| GMM-EN | 0.1932 | 0.1967 | 0.1949 | 0.0213 | 0.0227 | 0.0220 |
| PAE | 0.7276 | 0.7397 | 0.7336 | 0.1894 | 0.2062 | 0.1971 |
| E2E-AE | 0.0024 | 0.0025 | 0.0024 | 0.1064 | 0.1316 | 0.1176 |
| PAE-GMM-EM | 0.7183 | 0.7311 | 0.7246 | 0.4745 | 0.4538 | 0.4635 |
| PAE-GMM | 0.7251 | 0.7384 | 0.7317 | 0.4532 | **0.4881** | 0.4688 |
| DAGMM-p | 0.7579 | 0.7710 | 0.7644 | 0.4723 | 0.4725 | 0.4713 |
| DAGMM-NVI | 0.9290 | **0.9447** | 0.9368 | 0.4383 | 0.4587 | 0.4470 |
| DAGMM | **0.9297** | 0.9442 | **0.9369** | **0.4766** | 0.4834 | **0.4782** |

| Method | Arrhythmia | | | KDDCUP-Rev | | |
|---|---|---|---|---|---|---|
| | Precision | Recall | $F_1$ | Precision | Recall | $F_1$ |
| OC-SVM | **0.5397** | 0.4082 | 0.4581 | 0.7148 | **0.9940** | 0.8316 |
| DSEBM-r | 0.1515 | 0.1513 | 0.1510 | 0.2036 | 0.2036 | 0.2036 |
| DSEBM-e | 0.4667 | 0.4565 | 0.4601 | 0.2212 | 0.2213 | 0.2213 |
| DCN | 0.3758 | 0.3907 | 0.3815 | 0.2875 | 0.2895 | 0.2885 |
| GMM-EN | 0.3000 | 0.2792 | 0.2886 | 0.1846 | 0.1746 | 0.1795 |
| PAE | 0.4393 | 0.4437 | 0.4403 | 0.7835 | 0.7817 | 0.7826 |
| E2E-AE | 0.4667 | 0.4538 | 0.4591 | 0.7434 | 0.7463 | 0.7448 |
| PAE-GMM-EM | 0.3970 | 0.4168 | 0.4056 | 0.2822 | 0.2847 | 0.2835 |
| PAE-GMM | 0.4575 | 0.4823 | 0.4684 | 0.6307 | 0.6278 | 0.6292 |
| DAGMM-p | 0.4909 | 0.4679 | 0.4787 | 0.2750 | 0.2810 | 0.2780 |
| DAGMM-NVI | 0.5091 | 0.4892 | 0.4981 | 0.9211 | 0.9211 | 0.9211 |
| DAGMM | 0.4909 | **0.5078** | **0.4983** | **0.9370** | 0.9390 | **0.9380** |

Table 2: Average precision, recall, and $F_1$ from DAGMM and the baseline methods. For each metric, the best result is shown in bold.

In the second set of experiment, we investigate how DAGMM responds to contaminated training data. In each run, we reserve 50% of data by random sampling for testing. For the rest 50%, we take all samples from the normal class mixed with $c\%$ of samples from the anomaly class for model training.

| Ratio $c$ | DAGMM | | | DCN | | |
|---|---|---|---|---|---|---|
| | Precision | Recall | $F_1$ | Precision | Recall | $F_1$ |
| 1% | 0.9201 | 0.9337 | 0.9268 | 0.7585 | 0.7611 | 0.7598 |
| 2% | 0.9186 | 0.9340 | 0.9262 | 0.7380 | 0.7424 | 0.7402 |
| 3% | 0.9132 | 0.9272 | 0.9201 | 0.7163 | 0.7293 | 0.7228 |
| 4% | 0.8837 | 0.8989 | 0.8912 | 0.6971 | 0.7106 | 0.7037 |
| 5% | 0.8504 | 0.8643 | 0.8573 | 0.6763 | 0.6893 | 0.6827 |

| Ratio $c$ | DSEBM-e | | | OC-SVM | | |
|---|---|---|---|---|---|---|
| | Precision | Recall | $F_1$ | Precision | Recall | $F_1$ |
| 1% | 0.6995 | 0.7135 | 0.7065 | 0.7129 | 0.6785 | 0.6953 |
| 2% | 0.6780 | 0.6876 | 0.6827 | 0.6668 | 0.5207 | 0.5847 |
| 3% | 0.6213 | 0.6367 | 0.6289 | 0.6393 | 0.4470 | 0.5261 |
| 4% | 0.5704 | 0.5813 | 0.5758 | 0.5991 | 0.3719 | 0.4589 |
| 5% | 0.5345 | 0.5375 | 0.5360 | 0.1155 | 0.3369 | 0.1720 |

Table 3: Anomaly detection results on contaminated training data from KDDCUP

Table 3 reports the average precision, recall, and $F_1$ score after 20 runs of DAGMM, DCN, DSEBM-e, and OC-SVM on the KDDCUP dataset, respectively. As expected, contaminated training data negatively affect detection accuracy. When contamination ratio $c$ increases from 1% to 5%, average precision, recall, and $F_1$ score decrease for all the methods. Meanwhile, we notice that DAGMM is able to maintain good detection accuracy with 5% contaminated data. For OC-SVM, we adopt the same parameter setting used in the experiment with clean training data, and observe that OC-SVM is

more sensitive to contamination ratio. In order to receive better detection accuracy, it is important to train a model with high-quality data (*i.e.,* clean or keeping contamination ratio as low as possible).

In sum, the DAGMM learned by end-to-end training achieves the state-of-the-art accuracy on the public benchmark datasets, and provides a promising alternative for unsupervised anomaly detection.

## 4.5 VISUALIZATION ON THE LEARNED LOW-DIMENSIONAL REPRESENTATION

In this section, we use an example to demonstrate the advantage of DAGMM learned by end-to-end training, compared with the baselines that rely on pre-trained deep autoencoders.

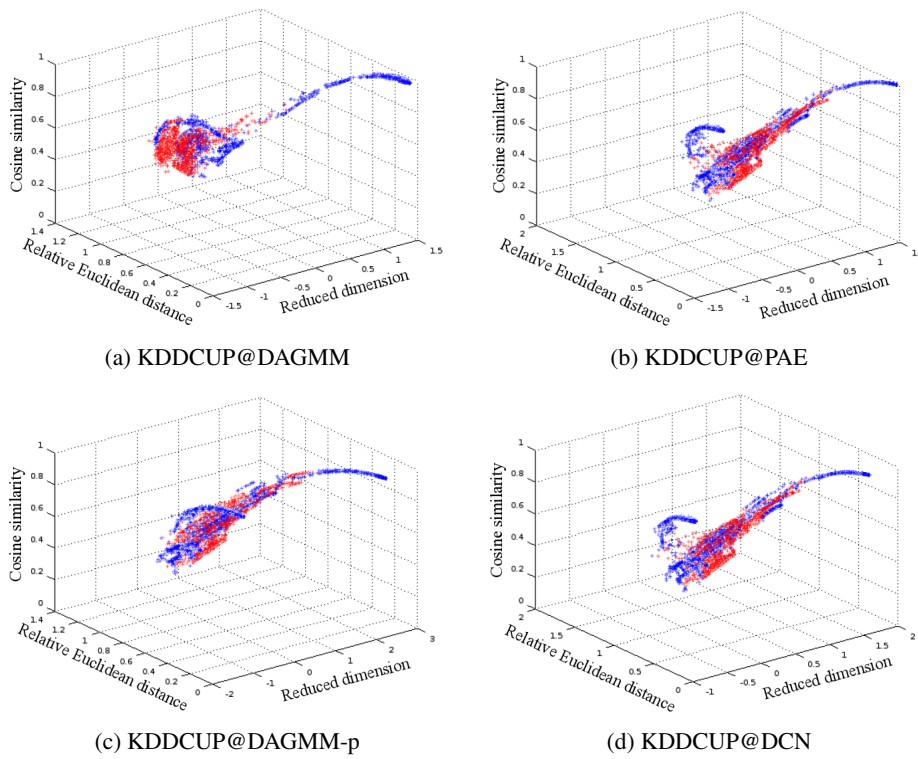

(a) KDDCUP@DAGMM

(b) KDDCUP@PAE

(c) KDDCUP@DAGMM-p

(d) KDDCUP@DCN

Figure 3: KDDCUP samples in the learned 3-dimensional space by DAGMM, PAE, DAGMM-p, and DCN, where red points are samples from anomaly class and blue ones are samples from normal class

Figure 3 shows the low-dimensional representation learned by DAGMM, PAE, DAGMM-p, and DCN, from one of the experiment runs on the KDDCUP dataset. First, we can see from Figure 3a that DAGMM can better separate anomalous samples from normal samples in the learned low-dimensional space, while anomalies overlap more with normal samples in the low-dimensional space learned by PAE, DAGMM-p, or DCN. Second, Even if DAGMM-p and DCN take effort to fine-tune the pre-trained deep autoencoder by its estimation network or k-means regularization, one could barely see significant change among Figure 3b, Figure 3c, and Figure 3d, where many anomalous samples are still mixed with normal samples. Indeed, when a deep autoencoder is pre-trained, it tends to be stuck in a good local optima for the purpose of reconstruction only, but it could be suboptimal for the subsequent density estimation tasks. In addition, in our study, we find that the reconstruction error in a trained DAGMM is as low as the error received from a pre-trained deep autoencoder (*e.g.,* around 0.26 in terms of per-sample reconstruction error for KDDCUP). Meanwhile, we also observe that it is difficult to reduce the reconstruction error for a deep autoencoder of the identical structure by end-to-end training (*e.g.,* around 1.13 in terms of per-sample reconstruction error for KDDCUP). In other words, the compression network and estimation network mutually boost each others' performance during end-to-end training: the regularization introduced by the estimation network helps the deep autoencoder escape from less attractive local optima for better compression, while the compression network feeds more meaningful low-dimensional representations to estimation network for robust

density estimation. In Appendix C, for readers of interest, we show the visualization of the latent representation learned by DSEBM.

In summary, our experimental results show that DAGMM suggests a promising direction for density estimation and anomaly detection, where one can combine the forces of dimensionality reduction and density estimation by end-to-end training.

In Appendix E, we provide another case study to discuss which kind of samples benefit more from joint training in DAGMM for readers of interest.

## 5 CONCLUSION

In this paper, we propose the Deep Autoencoding Gaussian Mixture Model (DAGMM) for unsupervised anomaly detection. DAGMM consists of two major components: compression network and estimation network, where the compression network projects samples into a low-dimensional space that preserves the key information for anomaly detection, and the estimation network evaluates sample energy in the low-dimensional space under the framework of Gaussian Mixture Modeling. DAGMM is friendly to end-to-end training: (1) the estimation network predicts sample mixture membership so that the parameters in GMM can be estimated without alternating procedures; and (2) the regularization introduced by the estimation network helps the compression network escape from less attractive local optima and achieve low reconstruction error by end-to-end training. Compared with the pre-training strategy, the end-to-end training could be more beneficial for density estimation tasks, as we can have more freedom to adjust dimensionality reduction processes to favor the subsequent density estimation tasks. In the experimental study, DAGMM demonstrates superior performance over state-of-the-art techniques on public benchmark datasets with up to $14\%$ improvement on the standard $F_1$ score, and suggests a promising direction for unsupervised anomaly detection on multi- or high-dimensional data.

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

# A    BASELINE CONFIGURATION

**OC-SVM**. Unlike other baselines that only need decision thresholds in the testing phase, OC-SVM needs parameter $\nu$ be set in the training phase. Although $\nu$ intuitively means anomaly ratio in training data, it is non-trivial to set a reasonable $\nu$ in the case where training data are all normal samples and anomaly ratio in the testing phase could be arbitrary. In this study, we simply perform exhaustive search to find the optimal $\nu$ that renders the highest $F_1$ score on individual datasets. In particular, $\nu$ is set to be $0.1$, $0.02$, $0.04$, and $0.1$ for KDDCUP, Thyroid, Arrhythmia, and KDDCUP-Rev, respectively.

**DSEBM**. We use the network structure for the encoding in DAGMM as guidelines to set up DSEBM instances. For KDDCUP and KDDCUP-Rev, it is configured as FC(120, 60, softplus)-FC(60, 30, softplus)-FC(30, 10, softplus)-FC(10, 1, softplus). For Thyroid, it is FC(6, 12, softplus)-FC(12, 4, softplus)-FC(4, 1, softplus). For Arrhythmia, it is FC(274, 10, softplus)-FC(10, 2, softplus). Moreover, for KDDCUP, Thyroid, Arrhythmia, and KDDCUP-Rev, the numbers of epochs are 200, 20000, 10000, and 400, respectively, and the sizes of mini-batches are 1024, 1024, 128, and 1024, respectively.

**DCN**. We use the network configuration for the autoencoder in DAGMM as guidelines to set up autoencoders in DCN. For KDDCUP and KDDCUP-Rev, the structure is FC(120, 60, $\tanh$)-FC(60, 30, $\tanh$)-FC(30, 10, $\tanh$)-FC(10, 1, none)-FC(1, 10, $\tanh$)-FC(10, 30, $\tanh$)-FC(30, 60, $\tanh$)-FC(60, 120, none). For Thyroid, it is FC(6, 12, $\tanh$)-FC(12, 4, $\tanh$)-FC(4, 1, none)-FC(1, 4, $\tanh$)-FC(4, 12, $\tanh$)-FC(12, 6, none). For Arrhythmia, it is FC(274, 10, $\tanh$)-FC(10, 2, none)-FC(2, 10, $\tanh$)-FC(10, 274, none). Moreover, for KDDCUP, Thyroid, Arrhythmia, and KDDCUP-Rev, the numbers of epochs for per-layer pre-training are 200, 20000, 10000, and 400, respectively, the numbers of epochs for fine tuning are 200, 20000, 10000, and 400, respectively, and the sizes of mini-batches in all the training phases are 1024, 1024, 128, and 1024, respectively.

**GMM-EN**. GMM-EN also borrows the wisdom from the network configurations in DAGMM. For KDDCUP, it is FC(120, 60, $\tanh$)-FC(60, 30, $\tanh$)-FC(30, 10, $\tanh$)-FC(10, 1, none)-FC(1, 10, $\tanh$)-Drop(0.5)-FC(10, 4, softmax). For Thyroid, it is FC(6, 12, $\tanh$)-FC(12, 4, $\tanh$)-FC(4, 1, none)-FC(1, 10, $\tanh$)-Drop(0.5)-FC(10, 2, softmax). For Arrhythmia, it is FC(274, 10, $\tanh$)-FC(10, 2, none)-FC(2, 10, $\tanh$)-Drop(0.5)-FC(10, 2, softmax). For KDDCUP-Rev, it is FC(120, 60, $\tanh$)-FC(60, 30, $\tanh$)-FC(30, 10, $\tanh$)-FC(10, 1, none)-FC(1, 10, $\tanh$)-Drop(0.5)-FC(10, 2, softmax). For KDDCUP, Thyroid, Arrhythmia, and KDDCUP-Rev, the numbers of epochs for training are 200, 20000, 10000, and 400, respectively, and the sizes of mini-batches are 1024, 1024, 128, and 1024, respectively.

**PAE**. PAE shares identical network structures with the autoencoder in DAGMM. For KDDCUP, Thyroid, Arrhythmia, and KDDCUP-Rev, the numbers of epochs for per-layer pre-training are 200, 20000, 10000, and 400, respectively, the numbers of epochs for fine tuning are 200, 20000, 10000, and 400, respectively, and the sizes of mini-batches in all the training phases are 1024, 1024, 128, and 1024, respectively.

**E2E-AE**. E2E-AE shares identical network structures with the autoencoder in DAGMM. For KDD-CUP, Thyroid, Arrhythmia, and KDDCUP-Rev, the numbers of epochs for end-to-end training are 200, 20000, 10000, and 400, respectively, and the sizes of mini-batches are 1024, 1024, 128, and 1024, respectively.

**PAE-GMM-EM**. PAE-GMM and DAGMM share identical network configurations. For KDDCUP, Thyroid, Arrhythmia, and KDDCUP-Rev, the numbers of epochs for per-layer pre-training are 200, 20000, 10000, and 400, respectively, the numbers of epochs for fine tuning are 200, 20000, 10000, and 400, respectively, and the sizes of mini-batches in all the training phases are 1024, 1024, 128, and 1024, respectively. For GMM learning, the EM algorithm stops when the maximum difference of the parameters between current iteration and its previous iteration is smaller than $10^{-6}$.

**PAE-GMM**. PAE-GMM and DAGMM share identical network configurations. For KDDCUP, Thyroid, Arrhythmia, and KDDCUP-Rev, the numbers of epochs for per-layer pre-training are 200, 20000, 10000, and 400, respectively, the numbers of epochs for fine tuning or GMM training are 200, 20000, 10000, and 400, respectively, and the sizes of mini-batches in all the training phases are 1024, 1024, 128, and 1024, respectively.

**DAGMM-p**. DAGMM-p and DAGMM share identical network configurations, but they are only different in training strategies: DAGMM adopts the strategy of end-to-end training, while DAGMM-p relies on pre-training to compression network and then joint fine-tuning. For KDDCUP, Thyroid, Arrhythmia, and KDDCUP-Rev, the numbers of epochs for per-layer pre-training are 200, 20000, 10000, and 400, respectively, the numbers of epochs for fine tuning are 200, 20000, 10000, and 400, respectively, and the sizes of mini-batches in all the training phases are 1024, 1024, 128, and 1024, respectively.

**DAGMM-NVI**. DAGMM and DAGMM-NVI share identical network configurations and training strategies as discussed in Section 4.

## B CUMULATIVE DISTRIBUTION FUNCTION OF THE ENERGY FUNCTION LEARNED BY DAGMM

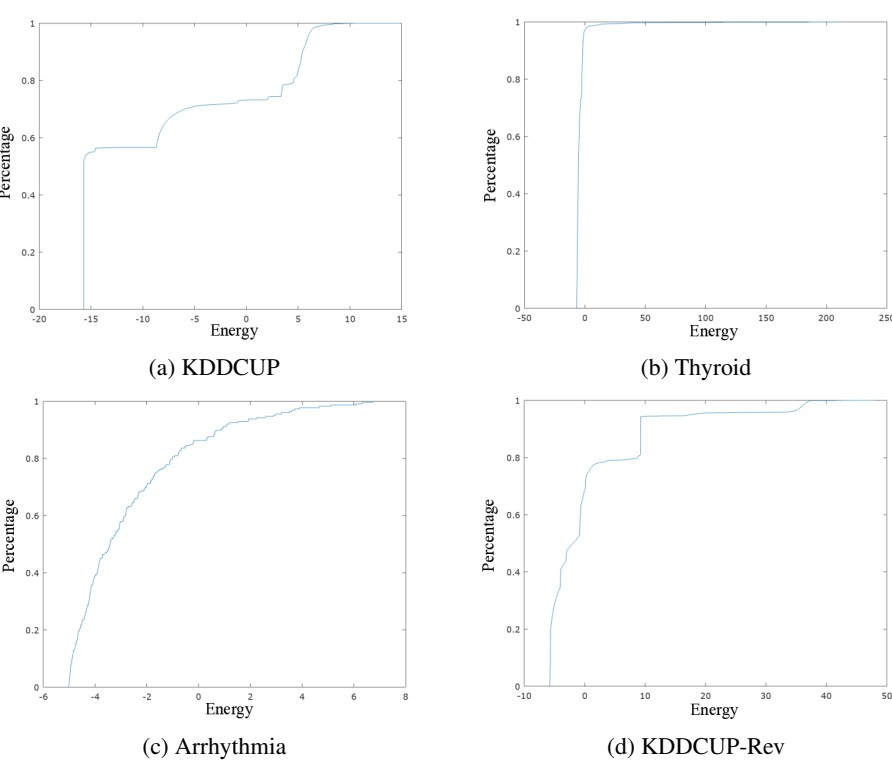

Figure 4: The cumulative distribution functions of the energy function are learned by DAGMM on KDDCUP, Arrhythmia, Thyroid, and KDDCUP-Rev, respectively. The horizontal axis denotes the energy space, and the vertical axis denotes the percentage.

Figure 4 shows the cumulative distribution function (cdf) of the energy function learned by DAGMM on KDDCUP, Arrhythmia, Thyroid, and KDDCUP-Rev, respectively. In particular, on KDDCUP and KDDCUP-Rev, we observe rapid energy increase at around $80\%$, and most samples whose energy is beyond the 80th percentile are true anomalous samples.

# C  Low-dimensional representation learned by DSEBM

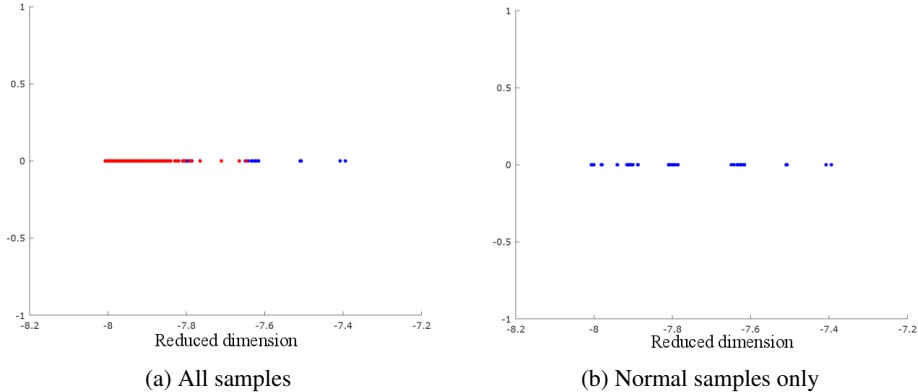

(a) All samples                    (b) Normal samples only

Figure 5: KDDCUP samples in the reduced 1-dimensional space by DSEBM, where red points are samples from the anomaly class and blue ones are samples from the normal class

Figure 5 demonstrates the reduced 1-dimensional representation for KDDCUP samples learned by DSEBM, where Figure 5a includes all the samples and Figure 5b includes the normal samples only. As shown above, normal and anomalous samples are mixed in the range of $[-8, -7.8]$. For samples in this range, it is difficult to use the energy derived from the latent representation to separate them.

# D  Reconstruction features in DAGMM

In this section, we detail the discussion on reconstruction features.

**Why reconstruction features are important?** We realize the importance of reconstruction features from our investigation on a private network security dataset. In this dataset, normal samples are normal network flows, and anomalies are network flows with spoofing attack. As it is difficult to analyze the samples from their original space with 20 dimensions, we utilize deep autoencoders to perform dimension reduction. In this case, we are a little bit ambitious, and reduce dimensions from 20 to 1. In the reduced 1-dimensional space, for some of the anomalies, we are able to easily separate them from normal samples. However, for the rest, their latent representations are quite similar to the representations of the normal samples. Meanwhile, in the original space, they are actually quite different from the normal ones. Inspired by this observation, we investigate their $L_2$ reconstruction error, and obtain the plot shown in Figure 1. In Figure 1, the red points in the top-right corner are the anomalies sharing similar representations with the normal samples in the reduced space. With the additional view from reconstruction error, it becomes easier to separate these anomalies from the normal samples. In our study, this concrete example motivates us to include reconstruction features into DAGMM.

**What are the guidelines for reconstruction feature selection?** In practice, one can select reconstruction features by the following rules. First, for an error metric used to derive a reconstruction feature, its analytical form should be continuous and differentiable. Second, the output of an error metric should be in a range of relatively small values for the ease of training the estimation network in DAGMM. In the experiment of this paper, we select cosine similarity and relative Euclidean distance based on these two rules. For cosine similarity, it is continuous and differentiable, and the range of its output is $[-1, 1]$. For relative Euclidean distance, it is also continuous and differentiable. Theoretically, the range of its output is $[0, +\infty)$. On the datasets considered in the experiment, we observe that its output is usually a small positive value; therefore, we include this metric as one of the reconstruction features.

In sum, as long as an error metric meets the above two rules, it could serve as a candidate metric to derive a reconstruction feature for DAGMM.

# E    CASE STUDY: WHEN JOINT TRAINING OUTPERFORMS DECOUPLED TRAINING?

In this section, we perform a case study to investigate what kind of samples benefit more from the joint training applied in DAGMM over decoupled training. In the evaluation, we employ PAE-GMM as a representative for the methods that leverage decoupled training, and the following results are generated from one run on the KDDCUP dataset.

|                | PAE-GMM detect | PAE-GMM miss |
|----------------|:--------------:|:------------:|
| DAGMM detect   | 34,285         | 12,038       |
| DAGMM miss     | 1,640          | 926          |

Table 4: The comparison of anomaly detection results between DAGMM and PAE-GMM

Table 4 presents the comparison between DAGMM and PAE-GMM in terms of their anomaly detection results. In the testing data of this run, there are $48, 889$ anomalies in total, where $34, 285$ of them are detected by both techniques, $926$ anomalies can be detected by neither of them, $1, 640$ can only be detected by PAE-GMM, and $12, 038$ of them can only be detected by DAGMM. Next, we drill deeper and investigate the commonalities of these $12, 038$ anomalies that can only be detected by DAGMM.

Figure 6 illustrates sample distribution in the low-dimensional spaces learned by DAGMM and PAE-GMM, where Figure 6a (6b) includes all the normal samples and anomalies, Figure 6c (6d) includes all the normal samples and the anomalies detected by both techniques, and Figure 6e (6f) includes all the normal samples and the anomalies detected by DAGMM only. From Figure 6c and 6d, we observe that the anomalies of low cosine similarity and high relative Euclidean distance could be the easy ones that are captured by both techniques. For the difficult ones shown in Figure 6e and 6f, we observe that they usually have medium level of relative Euclidean distance (in the range of $[1.0, 1.2]$ for both cases) with larger than $0.6$ cosine similarity. For such anomalous samples, the model learned by PAE-GMM has difficult time to separate them from the normal samples. In addition, we also observe that the model learned by DAGMM tends to assign lower cosine similarity to such anomalies than PAE-GMM does, which also makes it easier to differentiate the anomalies from the normal samples.

# F    HOW THE HYPERPARAMETERS IN THE OBJECTIVE FUNCTION IMPACT DAGMM

As shown in Equation (7), the objective function of DAGMM includes three components: the loss function from deep autoencoder, the energy function from estimation network, and the penalty function for covariance matrices. The coefficient ratio among the three components can be characterized as $1 : \lambda_1 : \lambda_2$. In terms of $\lambda_1$, a large value could make the loss function of deep autoencoder play little role in optimization so that we are unable to obtain a good reduced representation for input samples, while a small value could lead to ineffective estimation network so that GMM is not well trained. For $\lambda_2$ of a large value, DAGMM tends to find GMM with large covariance, which is less desirable as many samples will have high energy as rare events. For $\lambda_2$ of a small value, the regularization may not be strong enough to counter the singularity effect.

In our exploration, we find the ratio $1 : 0.1 : 0.005$ consistently delivers expected results across all the datasets in the experiment. To investigate the sensitivity of this ratio, we vary its base and see how different bases affect anomaly detection accuracy. For example, when the base is set to 2, $\lambda_1$ and $\lambda_2$ are adjusted to $0.2$ and $0.01$, respectively.

Table 5 shows the average precision, recall, and $F_1$ score after 20 runs of DAGMM on the KDDCUP dataset. As we vary the base from 1 to 9 with step 2, DAGMM performs in a consistent way, and $\lambda_1$, $\lambda_2$ are not sensitive to the changes on the base.

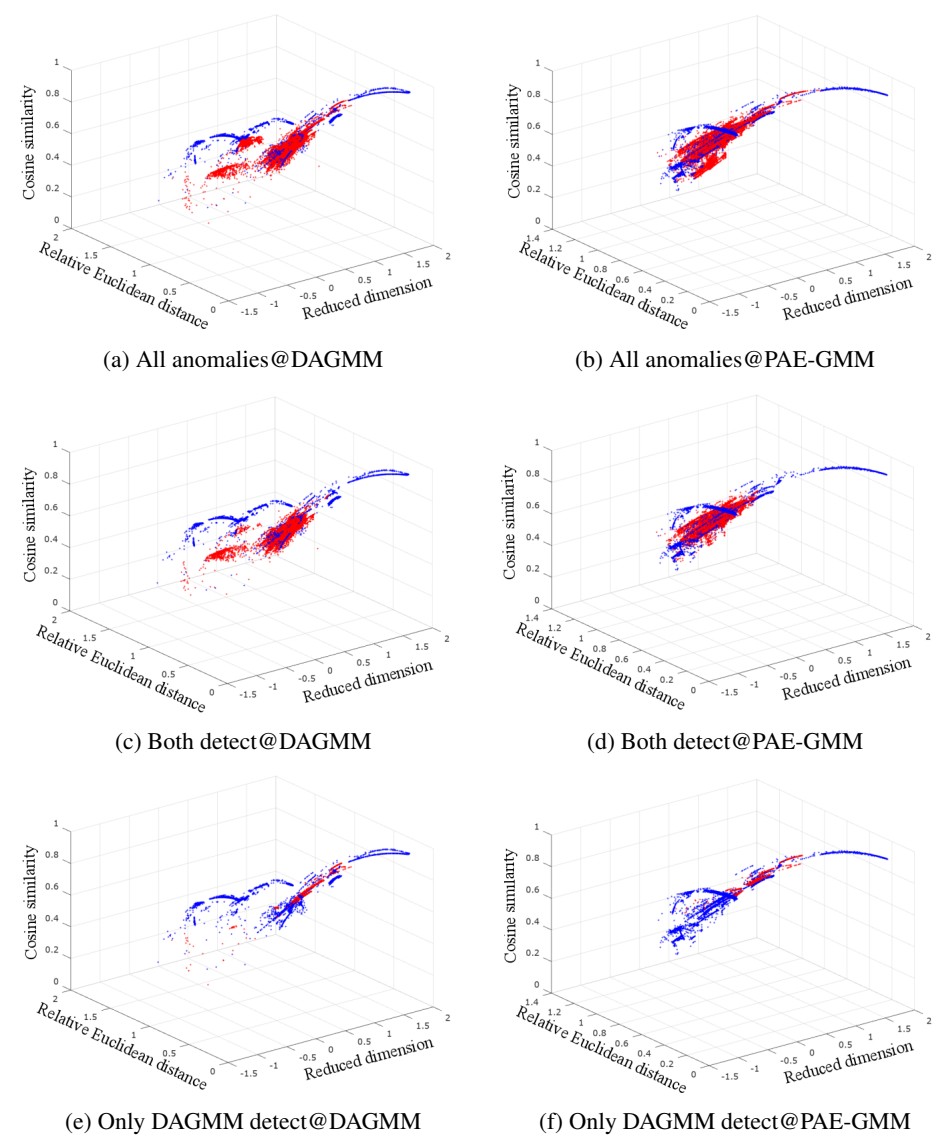

Figure 6: KDDCUP samples in the learned 3-dimensional space by DAGMM and PAE-GMM, where red points are samples from anomaly class and blue ones are samples from normal class

| Base | Precision | Recall | $F_1$ |
|------|-----------|--------|-------|
| 1 | 0.9298 | 0.9445 | 0.9371 |
| 3 | 0.9301 | 0.9442 | 0.9371 |
| 5 | 0.9296 | 0.9451 | 0.9373 |
| 7 | 0.9300 | 0.9453 | 0.9376 |
| 9 | 0.9300 | 0.9439 | 0.9369 |

Table 5: Sensitivity of $\lambda_1$ and $\lambda_2$ with fixed ratio $1 : 0.1 : 0.005$ on KDDCUP

