# OpenReview forum: "Deep Autoencoding Gaussian Mixture Model for Unsupervised Anomaly Detection"
_ICLR.cc/2018/Conference — Accept (Poster)_

### Official Review · AnonReviewer2 · 2017-11-26
**The paper presents a joint deep learning framework for dimension reduction-clustering, leads to competitive anomaly detection**

**Rating:** 8
**Confidence:** 5

**Review:**

1. This is a good paper, makes an interesting algorithmic contribution in the sense of joint clustering-dimension reduction for unsupervised anomaly detection
2. It demonstrates clear performance improvement via comprehensive comparison with state-of-the-art methods
3. Is the number of Gaussian Mixtures 'K' a hyper-parameter in the training process? can it be a trainable parameter?
4. Also, it will be interesting to get some insights or anecdotal evidence on how the joint learning helps beyond the decoupled learning framework, such as what kind of data points (normal and anomalous) are moving apart due to the joint learning

---

> ### Author Response · Authors · 2017-12-30
> **Response to the comments from the reviewer**
>
> Thanks for your valuable comments to our paper.
>
> Question 1. Is the number of Gaussian Mixtures 'K' a hyper-parameter in the training process? can it be a trainable parameter?
>
> Yes, in the current DAGMM, 'K' is a hyperparameter. In our opinion, it could be a trainable parameter. One possible way is to incorporate a Dirichlet Process prior into the optimization process so that an optimal 'K' can be automatically inferred. Meanwhile, one may have to make significant changes to the architecture of DAGMM, as the number of output neurons for the estimation network is tightly coupled with 'K'. A new architecture could be required in order to handle the uncertain 'K', if 'K' becomes trainable. In summary, we believe that the question of how to make 'K' trainable is interesting and important, and we will explore the answer to this question in our future work.
>
> Question 2. Also, it will be interesting to get some insights or anecdotal evidence on how the joint learning helps beyond the decoupled learning framework, such as what kind of data points (normal and anomalous) are moving apart due to the joint learning.
>
> Thanks for this constructive comment. In the revised paper, we added Appendix E, where we provide a case study on the KDDCUP dataset and show which kind of anomalous samples benefit more from joint learning.

---

### Official Review · AnonReviewer1 · 2017-11-27
**Clear paper on joint optimization of dimension reduction and density estimation, more work needed on the justification and the experimental setup.**

**Rating:** 8
**Confidence:** 4

**Review:**

The paper presents a new technique for anomaly detection where the dimension reduction and the density estimation steps are jointly optimized. The paper is rigorous and ideas are clearly stated. The idea to constraint the dimension reduction to fit a certain model, here a GMM, is relevant, and the paper provides a thorough comparison with recent state-of-the-art methods. My main concern is that the method is called unsupervised, but it uses the class information in the training, and also evaluation. I'm also not convinced of how well the Gaussian model fits the low-dimensional representation and how well can a neural network compute the GMM mixture memberships.

1. The framework uses the class information, i.e., “only data samples from the normal class are used for training”, but it is still considered unsupervised. Also, the anomaly detection in the evaluation step is based on a threshold which depends on the percentage of known anomalies, i.e., a priori information. I would like to see a plot of the sample energy as a function of the number of data points. Is there an elbow that indicates the threshold cut? Better yet it would be to use methods like Local Outlier Factor (LOF) (Breunig et al., 2000 – LOF:Identifying Density-based local outliers) to detect the outliers (these methods also have parameters to tune, sure, but using the known percentage of anomalies to find the threshold is not relevant in a purely unsupervised context when we don't know how many anomalies are in the data).
2. Is there a theoretical justification for computing the mixture memberships for the GMM using a neural network?
3. How do the regularization parameters \lambda_1 and \lambda_2 influence the results?
4. The idea to jointly optimize the dimension reduction and the clustering steps was used before neural nets (e.g., Yang et al., 2014 -  Unsupervised dimensionality reduction for Gaussian mixture model). Those approaches should at least be discussed in the related work, if not compared against.
5. The authors state that estimating the mixture memberships with a neural network for GMM in the estimation network instead of the standard EM algorithm works better. Could you provide a comparison with EM?
6. In the newly constructed space that consists of both the extracted features and the representation error, is a Gaussian model truly relevant? Does it well describe the new space? Do you normalize the features (the output of the dimension reduction and the representation error are quite different)? Fig. 3a doesn't seem to show that the output is a clear mixture of Gaussians.
7. The setup of the KDDCup seems a little bit weird, where the normal samples and anomalies are reversed (because of percentage), where the model is trained only on anomalies, and it detects normal samples as anomalies ... I'm not convinced that it is the best example, especially that is it the one having significantly better results, i.e. scores ~ 0.9 vs. scores ~0.4/0.5 score for the other datasets.
8. The authors mention that “we can clearly see from Fig. 3a that DAGMM is able to well separate ...” - it is not clear to me, it does look better than the other ones, but not clear. If there is a clear separation from a different view, show that one instead. We don't need the same view for all methods.
9. In the experiments the reduced dimension used is equal to 1 for two of the experiments and 2 for one of them. This seems very drastic!

Minor comments:

1. Fig.1: what dimension reduction did you use? Add axis labels.
2. “DAGMM preserves the key information of an input sample” - what does key information mean?
3. In Fig. 3 when plotting the results for KDDCup, I would have liked to see results for the best 4 methods from Table 1, OC-SVM performs better than PAE. Also DSEBM-e and DSEBM-r seems to perform very well when looking at the three measures combined. They are the best in terms of precision.
4. Is the error in Table 2 averaged over multiple runs? If yes, how many?

Quality – The paper is thoroughly written, and the ideas are clearly presented. It can be further improved as mentioned in the comments.

Clarity – The paper is very well written with clear statements, a pleasure to read.

Originality – Fairly original, but it still needs some work to justify it better.

Significance – Constraining the dimension reduction to fit a certain model is a relevant topic, but I'm not convinced of how well the Gaussian model fits the low-dimensional representation and how well can a neural network compute the GMM mixture memberships.

---

> ### Author Response · Authors · 2017-12-30
> **Response to the comments from the reviewer (Part 1)**
>
> Due to the length constraint on comment, we have to split our response into several parts.
>
> Thanks for your valuable comments to our paper.
>
> Question 1. The framework uses the class information, i.e., only data samples from the normal class are used for training, but it is still considered unsupervised. Also, the anomaly detection in the evaluation step is based on a threshold which depends on the percentage of known anomalies, i.e., a priori information. I would like to see a plot of the sample energy as a function of the number of data points. Is there an elbow that indicates the threshold cut? Better yet it would be to use methods like Local Outlier Factor (LOF) (Breunig et al., 2000 LOF:Identifying Density-based local outliers) to detect the outliers (these methods also have parameters to tune, sure, but using the known percentage of anomalies to find the threshold is not relevant in a purely unsupervised context when we don't know how many anomalies are in the data).
>
> We answer this question from three aspects.
>
> First, why do we only use samples from normal class for training?
>
> In general, there are two settings for model training in unsupervised anomaly detection: clean training data and contaminated training data.
>
> Clean training data is a widely adopted setting in many applications. For example, in the case of system fault detection, the detectors are trained by system logs generated from normal days. Under this setting, model training is still unsupervised, as there is no guidance from data labels that differentiate normal and abnormal samples. The experiment reported in Table 2 follows this setting.
>
> Contaminated training data is another setting, where we are not sure if there are anomaly data in the training data or not. Usually, the anomaly detection technique works well only when the contamination ratio is small. Again, model training in this setting is unsupervised, as the model does not know which samples are contaminated and receives no guidance from data labels as well. In the revised paper, we added Table 3 to show how DAGMM and its baselines respond to the contaminated training data from the KDDCUP dataset.
>
> Second, why do we use prior knowledge to set threshold?
>
> As far as we know, it is inevitable to decide a threshold for unsupervised anomaly detection techniques. Given the anomaly score of a sample, we still need to answer the question: should I report this sample as anomaly? With a threshold, we are able to answer this question. For different techniques, the threshold may be decided at different phases. For OC-SVM, the threshold is decided at the model training phase (i.e., the parameter nu). For LOF and DAGMM, the threshold is decided at the testing phase.
>
> While it is important to decide a threshold, the problem of how to find the optimal threshold is non-trivial in practice. In most cases, it is a process with exploration and exploitation: If the threshold is too high with high false positive rate, we decrease the threshold a bit; If the threshold is too low with low recall, we may increase the threshold.
>
> In this work, we do not intend to solve the problem of how to find the optimal threshold. For validation purpose, we assume DAGMM and its baselines have the prior knowledge to set their own optimal threshold so that we can fairly compare their performance.
>
> Third, in the revised paper, we added Appendix B that reports the cdf of the energy function learned by DAGMM for all the datasets.
>
> Question 2. Is there a theoretical justification for computing the mixture memberships for the GMM using a neural network?
>
> The estimation network in DAGMM is related to existing techniques such as neural variational inference [1] and adaptive mixture of experts [2]. In the revised paper, we added Section 3.5 to discuss how our technique connects to neural variational inference, and added a baseline that optimizes DAGMM under the framework of neural variational inference.
>
> Question 3. How do the regularization parameters \lambda_1 and \lambda_2 influence the results?
>
> In the revised paper, we added Appendix F to discuss how these hyperparameters impact the performance of DAGMM.
>
> Question 4. The idea to jointly optimize the dimension reduction and the clustering steps was used before neural nets (e.g., Yang et al., 2014 -  Unsupervised dimensionality reduction for Gaussian mixture model). Those approaches should at least be discussed in the related work, if not compared against.
>
> Thanks for sharing the related work. In the revised paper, we added the discussion on them in Section 2.
>
> Reference
>
> [1] Andriy Mnih and Karol Gregor. "Neural variational inference and learning in belief networks." arXiv preprint arXiv:1402.0030 (2014).
> [2] Robert A Jacobs, Michael I Jordan, Steven J Nowlan, and Geoffrey E Hinton. Adaptive mixtures of local experts. Neural computation, 3(1):79?87, 1991.

---

> > ### Comment · AnonReviewer1 · 2018-01-03
> > **Revised review - Response to the authors comments**
> >
> > Thank you for your detailed responses. I am happy with the comments, and have revised the grading of the paper.
> >
> > I would add a short description for clean vs. contamined model training in unsupervised anomaly detection (as in your responses to the comments) for the readers not familiar with the literature. It's good to see results with both types of training, added in Table 3.
> >
> > Rewrite to make more clear: “For DSEBM, while it works reasonably well on multiple datasets, DAGMM outperforms as both latent representation and reconstruction error are jointly considered in energy modeling.”  Possible: “DSEBM works reasonably well on multiple datasets, but DAGMM outperforms it, as DAGMM takes into account both the latent representation and the reconstruction error in the energy modeling.”
> >
> > Appendix B is very useful, and if the authors think it's helpful, you could switch the axis, where the energy is a function of the percentage; that would also help with interpreting the threshold cut. Also possibly show the y axis as a log if that helps the visualization.

---

> ### Author Response · Authors · 2017-12-30
> **Response to the comments from the reviewer (Part 2)**
>
> Question 5. The authors state that estimating the mixture memberships with a neural network for GMM in the estimation network instead of the standard EM algorithm works better. Could you provide a comparison with EM?
>
> Thanks for pointing out this confusion. In this paper, we have no intention to claim that the estimation network works better than the traditional EM algorithm. Instead, our point is that anomaly detection tasks benefit more from the joint training of dimension reduction and density estimation, compared with decoupled training. Meanwhile, it is indeed interesting to see how well the EM algorithm works with deep autoencoders. Therefore, in the revised paper, we added a baseline called PAE-GMM-EM, which uses the EM algorithm to learn GMM.
>
> Question 6. In the newly constructed space that consists of both the extracted features and the representation error, is a Gaussian model truly relevant? Does it well describe the new space? Do you normalize the features (the output of the dimension reduction and the representation error are quite different)? Fig. 3a doesn't seem to show that the output is a clear mixture of Gaussians.
>
> In this work, we do not assume the underlying distribution is Gaussian. Instead, we utilize a mixture of Gaussian distributions to approximate an unknown distribution. Informally speaking, any distribution can be well approximated by a finite number of Gaussian mixtures.
>
> In the current DAGMM, we do not perform any normalization on the output of dimension reduction and reconstruction features. Instead, we normalize input samples and carefully select reconstruction features (metrics) to keep the values in the low-dimensional space relatively small so that they are friendly to the estimation network training. In the revised paper, we added Appendix D to discuss reconstruction feature selection.
>
> Question 7. The setup of the KDDCup seems a little bit weird, where the normal samples and anomalies are reversed (because of percentage), where the model is trained only on anomalies, and it detects normal samples as anomalies ... I'm not convinced that it is the best example, especially that is it the one having significantly better results, i.e. scores ~ 0.9 vs. scores ~0.4/0.5 score for the other datasets.
>
> Thanks for the suggestion. In the revised paper, we added one more dataset KDDCUP-Rev, which is derived from the KDDCUP dataset. In this dataset, "normal" samples are the majority class, and "attack" samples are anomalies.
>
> Question 8. The authors mention that we can clearly see from Fig. 3a that DAGMM is able to well separate ... - it is not clear to me, it does look better than the other ones, but not clear. If there is a clear separation from a different view, show that one instead. We don't need the same view for all methods.
>
> In the revised paper, we modified the presentation in the second paragraph of Section 4.5 to make it more objective.
>
> Question 9. In the experiments the reduced dimension used is equal to 1 for two of the experiments and 2 for one of them. This seems very drastic!
>
> We are also surprised by the fact that we can use 1 or 2 reduced dimensions to achieve the state-of-the-art performance. We will share the source code on github upon the acceptance of this work so that more people are able to verify this discovery.

---

> ### Author Response · Authors · 2017-12-30
> **Response to the comments from the reviewer (Part 3)**
>
> Question 10. Fig.1: what dimension reduction did you use? Add axis labels.
>
> We utilize a deep autoencoder to perform dimension reduction on the dataset presented in Fig.1. In the revised paper, we added this missing information in the figure caption, and added axis labels to all the figures.
>
> Question 11. DAGMM preserves the key information of an input sample - what does key information mean?
>
> Thanks for pointing out the confusion. As stated in the paper, the key information means the features derived from both the reduced dimensions discovered by dimensionality reduction and the induced reconstruction error, which are important for anomaly detection tasks.
>
> Question 12. In Fig. 3 when plotting the results for KDDCup, I would have liked to see results for the best 4 methods from Table 1, OC-SVM performs better than PAE. Also DSEBM-e and DSEBM-r seems to perform very well when looking at the three measures combined. They are the best in terms of precision.
>
> For OC-SVM, we use the RBF kernel with an infinite number of dimensions in its kernel space, which is difficult to visualize and compare with Fig.3.
>
> For DSEBM-e and DSEBM-r, they share the same latent representation. Unlike the methods presented in Fig.3, DSEBM does not include reconstruction features in energy modeling; therefore, it is also difficult to compare its visualization results with the ones in Fig.3. In the revised paper, we added Appendix C that includes the visualization results of DSEBM on the KDDCUP dataset.
>
> Question 13. Is the error in Table 2 averaged over multiple runs? If yes, how many?
>
> As stated in the experiment section, Table 2 reports the average results over 20 runs. In the revised paper, we emphasized this information at the beginning of multiple paragraphs.
>
> At last, we sincerely appreciate your constructive and very detailed comments.

---

### Official Review · AnonReviewer3 · 2017-11-29
**Strong anomaly detection by end-to-end architecture that predicts GMM parameters with tailored loss function**

**Rating:** 8
**Confidence:** 4

**Review:**

Summary

This applications paper proposes using a deep neural architecture to do unsupervised anomaly detection by learning the parameters of a GMM end-to-end with reconstruction in a low-dimensional latent space. The algorithm employs a tailored loss function that involves reconstruction error on the latent space, penalties on degenerate parameters of the GMM, and an energy term to model the probability of observing the input samples.

The algorithm replaces the membership probabilities found in the E-step of EM for a GMM with the outputs of a subnetwork in the end-to-end architecture. The GMM parameters are updated with these estimated responsibilities as usual in the M-step during training.

The paper demonstrates improvements in a number of public datasets. Careful reporting of the tuning and hyperparameter choices renders these experiments repeatable, and hence a suitable improvement in the field. Well-designed ablation studies demonstrate the importance of the architectural choices made, which are generally well-motivated in intuitions about the nature of anomaly detection.

Criticisms

Based on the performance of GMM-EN, the reconstruction error features are crucial to the success of this method. Little to no detail about these features is included. Intuitively, the estimation network is given the latent code conditioned and some (probably highly redundant) information about the residual structure remaining to be modeled.

Since this is so important to the results, more analysis would be helpful. Why did the choices that were made in the paper yield this success? How do you recommend other researchers or practitioners selected from the large possible space of reconstruction features to get the best results?

Quality

This paper does not set out to produce a novel network architecture. Perhaps the biggest innovation is the use of reconstruction error features as input to a subnetwork that predicts the E-step output in EM for a GMM. This is interesting and novel enough in my opinion to warrant publication at ICLR, along with the strong performance and careful reporting of experimental design.

---

> ### Author Response · Authors · 2017-12-30
> **Response to the comments from the reviewer**
>
> Thanks for your valuable comments to our paper.
>
> For the reconstruction features used in the experiment, we report their details in the first paragraph of Section 4.3. In the revised paper, we added Appendix D to discuss why reconstruction features are important to anomaly detection and the principles that guide us to find candidate reconstruction features.
>
> For the question of how to find the set of reconstruction features that deliver the best results, it is important, but non-trivial. In our study, we discovered the two reconstruction features used in the experiment through a manual data exploration process. The principles in Appendix D are important guidelines for choosing candidate reconstruction metrics.

---

### Public Comment · (anonymous) · 2017-12-02
**Some related works are not cited**

This work obviously ignored several important related previous work from the speech recognition community. All these works use GMMs model features produced by a bottle-neck feature extractor (trained in a way sort of similar to an autoencoder), and jointly trained the feature extractor & GMMs. For example,

1. M.Paulik,"Lattice-based training of bottleneck feature extraction neural networks", Proc. Interspeech 2013.
which trains GMMs using EM and bottle-neck features using SGD in an interleaved fashion.

2. E. Variani, E. McDermott, and G. Heigold, "A Gaussian mixture model layer jointly optimized with discriminative features within a deep neural network architecture", ICASSP 2015.
3. C. Zhang and P.C. Woodland, "Joint optimisation of tandem systems using Gaussian mixture density neural network discriminative sequence training", ICASSP 2017.
These two papers jointly trained GMMs and their bottle-neck features using SGD and different criterion.

4. Z. Tuuske, M. Sundermeyer, R. Schluuter, and H. Ney, "Integrating Gaussian mixtures into deep neural networks: Softmax layer with hidden variables", ICASSP 2015.
5. Z. Tuuske, P. Golik, R. Schluuter, and H. Ney, "Speaker adaptive joint training of Gaussian mixture models and bottleneck features", ASRU 2015.
These two papers trained a log-linear mixture model (seen as an extension to softmax) together with the features.

The major difference between the neural network architectures in this paper and those cited above (esp. those in 2 & 3) is perhaps mainly whether to use a separate network to estimate the membership of the sample. It is not certain if such a membership estimation network is useful given sufficient computational power.

---

> ### Author Response · Authors · 2017-12-30
> **Response to the comments**
>
> Thanks for sharing these related work with us. It is interesting to read these related techniques from the speech recognition community.
>
> In our revised paper, we added a paragraph in Section 2 to discuss their connection and difference. The main message is as follows: Unlike your mentioned existing methods, we focus on unsupervised settings: DAGMM extracts useful features for anomaly detection through linear/non-linear dimensionality reduction realized by a deep autoencoder, and jointly learns their density under the GMM framework by mixture membership estimation, for which DAGMM can be viewed as a more powerful deep version of adaptive mixture of experts (Jacobs et al. (1991)) in combination with a deep autoencoder. More importantly, DAGMM combines induced reconstruction error and learned latent representation for unsupervised anomaly detection.

---

### Decision · Program_Chairs · 2018-01-29
**ICLR 2018 Conference Acceptance Decision**

**Decision:**

Accept (Poster)

**Comment:**

 + Empirically convincing and clearly explained application: a novel deep learning architecture and approach is shown to significantly outperform state-of-the-art in unsupervised anomaly detection.
 - No clear theoretical foundation and justification is provided for the approach
 - Connexion and differentiation from prior work on simulataneous learning representation and fitting a Gaussian mixture to it would deserve a much more thorough discussion / treatment.